# Avoiding degradation in deep feed-forward networks by phasing out skip-connections

## Abstract

A widely observed phenomenon in deep learning is the degradation problem: increasing the depth of a network leads to a decrease in performance on both test *and* training data. Novel architectures such as ResNets and Highway networks have addressed this issue by introducing various flavors of skip-connections or gating mechanisms. However, the degradation problem persists in the context of plain feed-forward networks. In this work we propose a simple method to address this issue. The proposed method poses the learning of weights in deep networks as a constrained optimization problem where the presence of skip-connections is penalized by Lagrange multipliers. This allows for skip-connections to be introduced during the early stages of training and subsequently phased out in a principled manner. We demonstrate the benefits of such an approach with experiments on MNIST, fashion-MNIST, CIFAR-10 and CIFAR-100 where the proposed method is shown to greatly decrease the degradation effect and is often competitive with ResNets.

## 1 Introduction

The *representation view* of deep learning suggests that neural networks learn an increasingly abstract representation of input data in a hierarchical fashion (Zeiler & Fergus, 2014; Goodfellow et al., 2016; Greff et al., 2016). Such representations may then be exploited to perform various tasks such as image classification, machine translation and speech recognition.

A natural conclusion of the representation view is that deeper networks will learn more detailed and abstract representations as a result of their increased capacity. However, in the case of feed-forward networks it has been observed that performance deteriorates beyond a certain depth, even when the network is applied to training data. Recently, Residual Networks (ResNets; He et al. 2016a) and Highway Networks (Srivastava et al., 2015) have demonstrated that introducing various flavors of skip-connections or gating mechanisms makes it possible to train increasingly deep networks. However, the aforementioned degradation problem persists in the case of plain deep networks (i.e., networks without skip-connections of some form).

A widely held hypothesis explaining the success of ResNets is that the introduction of skip-connections serves to improve the conditioning of the optimization manifold as well as the statistical properties of gradients employed during training. Raiko et al. (2012) and Schraudolph (2012) show that the introduction of specially designed skip-connections serves to diagonalize the Fisher information matrix, thereby bringing standard gradient steps closer to the natural gradient. More recently, Balduzzi et al. (2017) demonstrated that the introduction of skip-connections helps retain the correlation structure across gradients. This is contrary to the gradients of deep feed-forward networks, which resemble white noise. More generally, the skip-connections are thought to reduce the effects of vanishing gradients by introducing a linear term (He et al., 2016b).

The goal of this work is to address the degradation issue in plain feed-forward networks by leveraging some of the desirable optimization properties of ResNets. We approach the task of learning parameters for a deep network under the framework of constrained optimization. This strategy allows us to introduce skip-connections penalized by Lagrange multipliers into the architecture of our network. In our setting, skip-connections play an important role during the initial training of the network and are subsequently removed in a principled manner. Throughout a series of experiments

we demonstrate that such an approach leads to improvements in generalization error when compared to architectures without skip-connections and is competitive with ResNets in some cases.

The contributions of this work are as follows:

- We propose alternative training strategy for plain feed-forward networks which reduces the degradation in performance as the depth of the network increases. The proposed method introduces skip-connections which are penalized by Lagrange multipliers. This allows for the presence of skip-connections to be iteratively phased out during training in a principled manner. The proposed method is thereby able to enjoy the optimization benefits associated with skip-connections during the early stages of training.
- A number of benchmark datasets are used to demonstrate the empirical capabilities of the proposed method. In particular, the proposed method greatly reduces the degradation effect compared to plain networks and is on several occasions competitive with ResNets.

## 2    RELATED WORK

The hierarchical nature of many feed-forward networks is loosely inspired by the structure of the visual cortex where neurons in early layers capture simple features (e.g., edges) which are subsequently aggregated in deeper layers (Hubel & Wiesel, 1962). This interpretation of neural networks suggests that the depth of a network should be maximized, thereby allowing the network to learn more abstract (and hopefully useful) representations (Bengio et al., 2013). However, a widely reported phenomenon is that deeper networks are more difficult to train. This is often termed the degradation effect in deep networks (Srivastava et al., 2015; He et al., 2016a). This effect has been partially attributed to optimization challenges such as vanishing and shattered gradients (Hochreiter et al., 2001; Balduzzi et al., 2017).

In the past these challenges have been partially addressed via the use of supervised and unsupervised pre-training (Bengio et al., 2009) and more recently through careful parameter initialization (Glorot & Bengio, 2010; He et al., 2015) and batch normalization (Ioffe & Szegedy, 2015). In the past couple of years further improvements have been obtained via the introduction of skip-connections. ResNets (He et al., 2016a;b) introduce residual blocks consisting of a residual function $\mathcal{F}$ together with a skip-connection. Formally, the residual block is defined as:

$$\mathbf{x}_{l+1} = \mathcal{F}_l(\mathbf{x}_l, \mathbf{W}_l) + \mathbf{W}_l' \mathbf{x}_l \tag{1}$$

where $\mathcal{F}_l : \mathbb{R}^n \to \mathbb{R}^{n'}$ represents some combination of affine transformation, non-linearity and batch normalization parameterized by $\mathbf{W}_l$. The matrix $\mathbf{W}_l'$ parameterizes a linear projection to ensure the dimensions are aligned[1]. More generally, ResNets are closely related to Highway Networks (Srivastava et al., 2015) where the output of each layer is defined as:

$$\mathbf{x}_{l+1} = \mathcal{F}_l(\mathbf{x}_l, \mathbf{W}_l) \cdot \mathcal{T}(\mathbf{x}_l, \mathbf{H}_l) + \mathbf{x}_l \cdot (1 - \mathcal{T}(\mathbf{x}_l, \mathbf{H}_l)), \tag{2}$$

where $\cdot$ denotes element-wise multiplication. In Highway Networks the output of each layer is determined by a gating function

$$\mathcal{T}(\mathbf{x}_l, \mathbf{H}_l) = \text{sigmoid}\,(\mathbf{H}_l \mathbf{x}_l)$$

inspired from LSTMs. We note that both ResNets and Highway Networks were introduced with the explicit goal of training deeper networks. Inspired by the success of the these methods, many variations have been proposed. Huang et al. (2016a) propose DenseNet, where skip-connections are passed from all previous activations. Huang et al. (2016b) propose to shorten networks during training by randomly dropping entire layers, leading to better gradient flow and information propagation, while using the full network at test time.

Recently, the goal of learning deep networks without skip-connections has begun to receive more attention. Zagoruyko & Komodakis (2017) propose a novel re-parameterization of weights in feed-forward networks which they call the Dirac parameterization. Instead of explicitly adding a skip-connection, they model the weights as a residual of the Dirac function, effectively moving the skip-connection inside the non-linearity. In related work, Balduzzi et al. (2017) propose to initialize

---

[1] Unless stated otherwise we will assume $\mathcal{F}$ retains the dimension of $\mathbf{x}_l$ and set $\mathbf{W}_l'$ to the identity.

weights in a CReLU activation function in order to preserve linearity during the initial phases of training. This is achieved by initializing the weights in a mirrored block structure. During training the weights are allowed to diverge, resulting in non-linear activations.

Finally, we note that while the aforementioned approaches have sought to train deeper networks via modifications to the network architecture (i.e., by adding skip-connections) success has also been obtained by modifying the non-linearities (Clevert et al., 2015; Klambauer et al., 2017).

## 3 VARIABLE ACTIVATION NETWORKS

The goal of this work is to train deep feed-forward networks without suffering from the degradation problem described in previous sections. To set notation, we denote $\mathbf{x}_0$ as the input and $\mathbf{x}_L$ as the output of a feed-forward network with $L$ layers. Given training data $\{\mathbf{y}, \mathbf{x}_0\}$ it is possible to learn parameters $\{\mathbf{W}_l\}_{l=1}^L$ by locally minimizing some objective function

$$\{\hat{\mathbf{W}}_l\}_{l=1}^L = \arg\min \mathcal{C}\left(\mathbf{y}, \mathbf{x}_L; \{\mathbf{W}_l\}_{l=1}^L\right). \tag{3}$$

First-order methods are typically employed due to the complexity of the objective function in equation (3). However, directly minimizing the objective is not practical in the context of deep networks: beyond a certain depth performance quickly deteriorates on both test *and* training data. Such a phenomenon does not occur in the presence of skip-connections. Accordingly, we take inspiration from ResNets and propose to modify equation (1) in the following manner[2]:

$$\mathbf{x}_{l+1} = \mathcal{F}_l(\mathbf{x}_l, \mathbf{W}_l) + (1 - \boldsymbol{\alpha}_l) \cdot \mathbf{x}_l \tag{4}$$

where $\boldsymbol{\alpha}_l \in [0, 1]^n$ determines the weighting given to the skip-connection. More specifically, $\boldsymbol{\alpha}_l$ is a vector were the entry $i$ dictates the presence and magnitude of a skip-connection for neuron $i$ in layer $l$. Due to the variable nature of parameters $\boldsymbol{\alpha}_l$ in equation (4), we refer to networks employing such residual blocks as Variable Activation Networks (VAN).

The objective of the proposed method is to train a feed-forward network under the constraint that $\boldsymbol{\alpha}_l = \mathbf{1}$ for all layers, $l$. When the constraint is satisfied all skip-connections are removed. The advantage of such a strategy is that we only require $\boldsymbol{\alpha}_l = \mathbf{1}$ at the *end* of training. This allows us to initialize $\boldsymbol{\alpha}_l$ to some other value, thereby relaxing the optimization problem and obtaining the advantages associated with ResNets during the early stages of training. In particular, whenever $\boldsymbol{\alpha}_l \neq \mathbf{1}$ information is allowed to flow through the skip-connections, alleviating issues associated with shattered and vanishing gradients.

As a result of the equality constraint on $\boldsymbol{\alpha}_l$, the proposed activation function effectively does not introduce any additional parameters. All remaining weights can be trained by solving the following constrained optimization problem:

$$\{\hat{\mathbf{W}}_l\}_{l=1}^L = \operatorname{argmin} \mathcal{C}\left(\mathbf{y}, \mathbf{x}_L; \{\mathbf{W}_l, \boldsymbol{\alpha}_l\}_{l=1}^L\right) \text{ such that } \boldsymbol{\alpha}_l = \mathbf{1} \text{ for } l = 1, \dots, L. \tag{5}$$

The associated Lagrangian takes the following simple form (Boyd & Vandenberghe, 2004):

$$\mathcal{L} = \mathcal{C}\left(\mathbf{y}, \mathbf{x}_L; \{\mathbf{W}_l, \boldsymbol{\alpha}_l\}_{l=1}^L\right) + \sum_{l=1}^L \boldsymbol{\lambda}_l^T(\boldsymbol{\alpha}_l - \mathbf{1}), \tag{6}$$

where each $\boldsymbol{\lambda}_l \in \mathbb{R}^n$ are the Lagrange multipliers associated with the constraints on $\boldsymbol{\alpha}_l$. In practice, we iteratively update $\boldsymbol{\alpha}_l$ via stochastic gradients descent (SGD) steps of the form:

$$\boldsymbol{\alpha}_l \leftarrow \boldsymbol{\alpha}_l - \eta \left(\frac{\partial \mathcal{C}}{\partial \boldsymbol{\alpha}_l} + \boldsymbol{\lambda}_l\right) \tag{7}$$

where $\eta$ is the step-size parameter for SGD. Throughout the experiments we will often take the non-linearity in $\mathcal{F}_l$ to be ReLU. Although not strictly required, we clip the values $\boldsymbol{\alpha}_l$ to ensure they remain in the interval $[0, 1]^n$.

---

[2]Our original formulation consisted of a convex sum over $\mathcal{F}$ and $\mathbf{x}_l$. However, we found this approach to be the most successful empirically. Appendix A provides a comparison to original formulations as well as some theoretical justifications for this improvement in performance.

From equation (6), we have that the gradients with respect to Lagrange multipliers are of the form:

$$\boldsymbol{\lambda}_l \leftarrow \boldsymbol{\lambda}_l + \eta' \left( \boldsymbol{\alpha}_l - \mathbf{1} \right), \tag{8}$$

We note that since we require $\boldsymbol{\alpha}_l \in [0,1]^n$, the values of $\boldsymbol{\lambda}_l$ are monotonically decreasing. As the value of Lagrange multiplier decreases, this in turn pushes $\boldsymbol{\alpha}_l$ towards $\mathbf{1}$ in equation (7). We set the step-size for the Lagrange multipliers, $\eta'$, to be a fraction of $\eta$. The motivation behind such a choice is to allow the network to adjust as we enforce the constraint on $\boldsymbol{\alpha}_l$.

## 4 EXPERIMENTS

The purpose of the experiments presented in this section is to demonstrate that the proposed method serves to effectively alleviate the degradation problem in deep networks. We first demonstrate the capabilities of the proposed method using a simple, non-convolutional architecture on the MNIST and Fashion-MNIST datasets (Xiao et al., 2017) in Section 4.1. More extensive comparisons are then considered on the CIFAR datasets (Krizhevsky & Hinton, 2009) in Section 4.2.

### 4.1 MNIST AND FASHION-MNIST

Networks of varying depths were trained on both MNIST and Fashion-MNIST datasets. Following Srivastava et al. (2015) the networks employed in this section were *thin*, with each layer containing 50 hidden units. In all networks the first layer was a fully connected plain layer followed by $l$ layers or residual blocks (depending on the architecture) and a final softmax layer. The proposed method is benchmarked against several popular architectures such as ResNets and Highway Networks as well as the recently proposed DiracNets (Zagoruyko & Komodakis, 2017). Plain networks without skip-connections are also considered. Finally, we also considered VAN network where the constraint $\boldsymbol{\alpha}_l = \mathbf{1}$ was not enforced. This corresponds to the case where $\boldsymbol{\lambda}_l = 0$ for all $l$. This comparison is included in order to study the capacity and flexibility of VAN networks without the need to satisfy the constraint to remove skip-connections. For clarity, refer to such networks as VAN ($\lambda = 0$) networks.

For all architectures the ReLU activation function was employed together with batch-normalization. In the case of ResNets and VAN, the residual function consisted of batch-normalization followed by ReLU and a linear projection.

The depth of the network varied from $l = 1$ to $l = 30$ hidden layers. All networks were trained using SGD with momentum. The learning rate is fixed at $\eta = 0.001$ and the momentum parameter at 0.9. Training consisted of 50 epochs with a batch-size of 128. In the case of VAN networks the $\boldsymbol{\alpha}_l$ values were initialized to 0 for all layers. As such, during the initial stages of training VAN networks where equivalent to ResNets. The step-size parameter for Lagrange multipliers, $\eta'$, was set to be one half of the SGD step-size, $\eta$. Finally, all Lagrange multipliers, $\boldsymbol{\lambda}_l$, are initialized to -1.

### RESULTS

The results are shown in Figure 1 where the test accuracy is shown as a function of the network depth for both the MNIST and Fashion-MNIST datasets. In both cases we see clear evidence of the degradation effect: the performance of plain networks deteriorates significantly once the network depth exceeds some critical value (approximately 10 layers). As would be expected, this is not the case for ResNets, Highway Networks and DiracNets as such architectures have been explicitly designed to avoid this behavior. We note that VAN networks do not suffer such a pronounced degradation as the depth increases. This provides evidence that the gradual removal of skip-connections via Lagrange multipliers leads to improved generalization performance compared to plain networks. Finally, we note that VAN networks obtain competitive results across all depths. Crucially, we note that VAN networks outperform plain networks across all depths, suggesting that the introduction of variable skip-connections may lead to convergence at local optima with better generalization performance. Finally, we note that VAN ($\lambda = 0$) networks, where no constraint is placed on skip-connections, obtain competitive results across all depths.

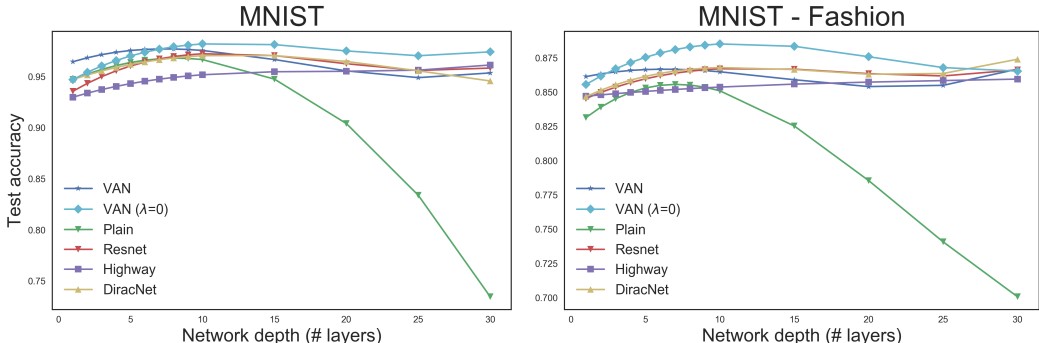

Figure 1: Results on MNIST (left) and fashion-MNIST (right) for various different architectures as the depth of the network varies from 1 to 30. Mean average test accuracy over 10 independent training sessions is shown. We note that with the exception of plain networks, the performance of all remaining architectures is stable as the number of layers increases.

## 4.2 CIFAR

As a more challenging benchmark we consider the CIFAR-10 and CIFAR-100 datasets. These consist of 60000 $32\times32$ pixel color images with 10 and 100 classes respectively. The datasets are divided into 50000 training images and 10000 test images.

We follow He et al. (2016a) and train deep convolutional networks consisting of four blocks each consisting of $n$ residual layers. The residual function is of the form `conv-BN-ReLU-conv-BN-ReLU`. This corresponds to the *pre-activation* function (He et al., 2016b). The convolutional layers consist of $3 \times 3$ filters with downsampling at the beginning of blocks 2, 3 and 4. The network ends with a fully connected softmax layer, resulting in a depth of $8n + 2$. The architecture is described in Table 1.

Networks were trained using SGD with momentum over 165 epochs. The learning rate was set to $\eta = 0.1$ and divided by 10 at the 82nd and 125th epoch. The momentum parameter was set to 0.9. Networks were trained using mini-batches of size 128. Data augmentation followed Lee et al. (2015): this involved random cropping and horizontal flips. Weights were initialized following He et al. (2015). As in Section 4.1, we initialize $\boldsymbol{\alpha}_l = 0$ for all layers. Furthermore, we set the step-size parameter for the Lagrange multipliers, $\eta'$, to be one tenth of $\eta$ and all Lagrange multipliers, $\boldsymbol{\lambda}_l$, are initialized to -1. On CIFAR-10 we ran experiments with $n \in \{1, 2, 3, 4, 5, 6, 8, 10\}$ yielding networks with depths ranging from 10 to 82. For CIFAR-100 experiments were run with $n \in \{1, 2, 3, 4\}$.

Table 1: Architecture of varying depth employed on CIFAR-10 and CIFAR-100 datasets. Downsampling occurs at the beginning of block 2, 3 and 4 with a stride of two.

| Layer name | Output size | Convolution |
|---|---|---|
| conv1 | $32 \times 32$ | $3 \times 3, 64$ |
| Block 1 | $32 \times 32$ | $\begin{bmatrix} 3\times3,64 \\ 3\times3,64 \end{bmatrix} \times n$ |
| Block 2 | $16 \times 16$ | $\begin{bmatrix} 3\times3,128 \\ 3\times3,128 \end{bmatrix} \times n$ |
| Block 3 | $8 \times 8$ | $\begin{bmatrix} 3\times3,256 \\ 3\times3,256 \end{bmatrix} \times n$ |
| Block 4 | $4 \times 4$ | $\begin{bmatrix} 3\times3,512 \\ 3\times3,512 \end{bmatrix} \times n$ |

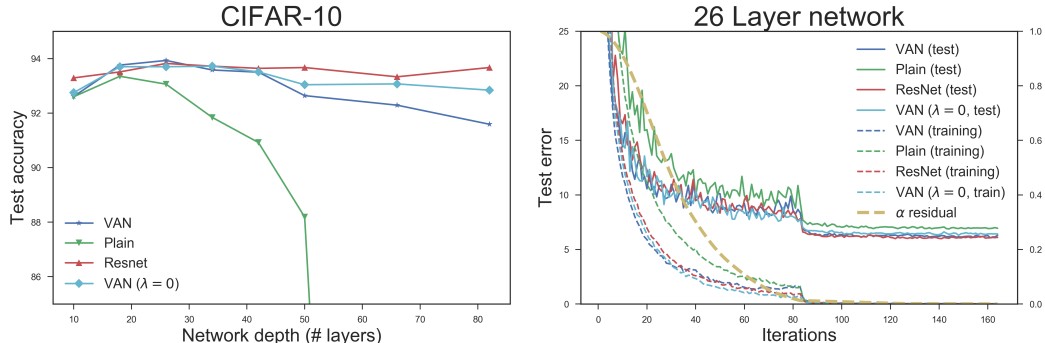

Figure 2: **Left:** Results on CIFAR-10 dataset are shown as the depth of networks increase. We note that the performance of both VAN and plain networks deteriorates as the depth increases, but the effect is far less pronounced for VAN networks. **Right:** Training and test error curves are shown for networks with 26 layers. We also plot the mean $\alpha$ residuals: $\frac{1}{L}\sum_{l=1}^{L}(\mathbf{1}-\boldsymbol{\alpha}_l)^2$ on the right axis.

RESULTS

Results for experiments on CIFAR-10 are shown in Figure 2. The left panel shows the mean test accuracy over five independent training sessions for ResNets, VAN, VAN ($\lambda = 0$) and plain networks. While plain networks provide competitive results for networks with fewer than 30 layers, their performance quickly deteriorates thereafter. We note that a similar phenomenon is observed in VAN networks but the effect is not as dramatic. In particular, the performance of VANs is similar to ResNets for networks with up to 40 layers. Beyond this depth, ResNets outperform VAN by an increasing margin. This holds true for both VAN and VAN ($\lambda = 0$) networks, however, the difference is reduced in magnitude in the case of VAN ($\lambda = 0$) networks. These results are in line with He et al. (2016b), who argue that scalar modulated skip-connections (as is the case in VANs where the scalar is $\mathbf{1} - \boldsymbol{\alpha}_l$) will either vanish or explode in very deep networks whenever the scalar is not the identity.

The right panel of Figure 2 shows the training and test error for a 26 layer network. We note that throughout all iterations, both the test and train accuracy of the VAN network dominates that of the plain network. The thick gold line indicates the mean residuals of the $\boldsymbol{\alpha}_l$ parameters across all layers. This is defined as $\frac{1}{L}\sum_{l=1}^{L}(\mathbf{1}-\boldsymbol{\alpha}_l)^2$ and is a measure of the extent to which skip-connections are present in the network. Recall that if all $\boldsymbol{\alpha}_l$ values are set to one then all skip-connections are removed (see equation (4)). From Figure 2, it follows that skip-connections are fully removed from the VAN network at approximately the $120^{th}$ iteration. More detailed traces of Lagrange multipliers and $\boldsymbol{\alpha}_l$ are provided in Appendix B.

A comparison of the performance of VAN networks in provided in Table 2. We note that while VAN networks do not outperform ResNets, they do outperform other alternatives such as Highway networks and FitNets (Romero et al., 2014) when networks of similar depths considered. However, it is important to note that both Highway networks and FitNets did not employ batch-normalization, which is a strong regularizer. In the case of both VAN and VAN ($\lambda = 0$) networks, the best performance is obtained with networks of 26 layers while ResNets continue to improve their performance as depth increases. Finally, current state-of-the-art performance, obtained by Wide ResNets (Zagoruyko & Komodakis, 2016) and DenseNet Huang et al. (2016a), are also provided in Table 2

Figure 3 provides results on the CIFAR-100 dataset. This dataset is considerably more challenging as it consists of a larger number of classes as well as fewer examples per class. As in the case of CIFAR-10, we observe a fall in the performance of both VAN and plain networks beyond a certain depth; in this case approximately 20 layers for plain networks and 30 layers for VANs. Despite this drop in performance, Table 2 indicates that the performance of VAN networks with both 18 and 26 layers are competitive with many alternatives proposed in the literature. Furthermore, we note that the performance of VAN ($\lambda = 0$) networks is competitive with ResNets in the context of the CIFAR-100 dataset.

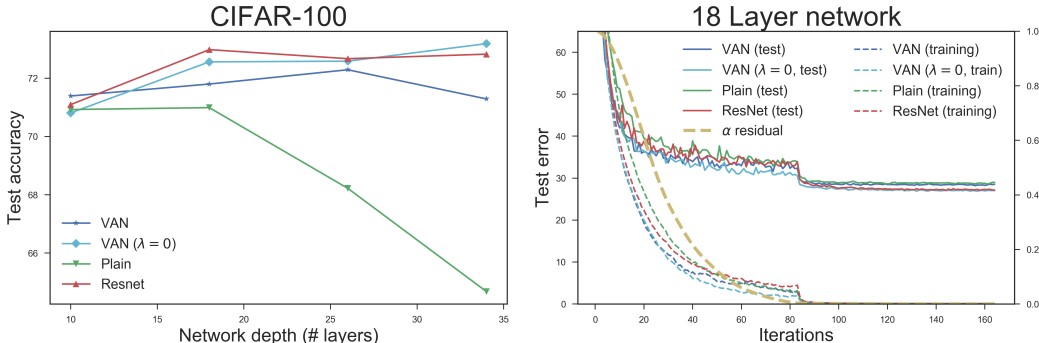

Figure 3: **Left:** Results on CIFAR-100 dataset are shown as the depth increases from 10 to 34 layers. We note that the performance of both VAN and plain networks deteriorates as the depth increases, but the effect is far less pronounced for plain networks. **Right:** Training and test error curves are shown for VAN and plain networks with 18 layers. The mean $\alpha$ residuals, $\frac{1}{L}\sum_{l=1}^{L}(\mathbf{1}-\boldsymbol{\alpha}_l)^2$, are shown in gold along the right axis.

Training curves are shown on the right hand side of Figure 3. As in the equivalent plot for CIFAR-10, the introduction and subsequent removal of skip-connections during training leads to improvements in generalization error.

Table 2: Comparison of VAN networks to results of other convolutional networks on CIFAR-10 and CIFAR-100. For VAN networks we report the best value as well as the mean and standard deviation over five independent training runs (in brackets). Results are ordered by performance on CIFAR-10. We add a $*$ to denote results which did not employ batch-normalization.

| Architecture | # Layers | CIFAR-10 (test error %) | CIFAR-100 (test error %) |
|---|---|---|---|
| Highway Network$^*$ | 32 | 8.80 | - |
| FitNet$^*$ | 19 | 8.39 | 35.04 |
| Highway Network$^*$ | 19 | 7.54 | 32.39 |
| DiracNet (width-1) | 34 | 7.10 | - |
| ELU$^*$ | 18 | 6.55 | 24.28 |
| VAN ($\lambda = 0$) | 26 | 6.29 ($6.40 \pm 0.16$) | 27.04 ($27.42 \pm 0.26$) |
| VAN ($\lambda = 0$) | 34 | 6.28 ($6.45 \pm 0.14$) | 26.46 ($26.81 \pm 0.31$) |
| VAN | 18 | 6.23 ($6.49 \pm 0.16$) | 28.20 ($28.42 \pm 0.36$) |
| VAN | 26 | 6.08 ($6.35 \pm 0.21$) | 27.70 ($28.01 \pm 0.39$) |
| DiracNet (width-2) | 34 | 5.60 | 26.72 |
| ResNet | 164 | 5.46 | 24.33 |
| Wide ResNet (width-10) | 28 | 4.00 | 19.25 |
| DenseNet | 160 | 3.46 | 17.18 |

## 5 DISCUSSION

This manuscript presents a simple method for training deep feed-forward networks which greatly reduces the degradation problem. In the past, the degradation issue has been successfully addressed via the introduction of skip-connections. As such, the goal of this work is to propose a new training regime which retains the optimization benefits associated with ResNets while ultimately phasing out skip-connections. This is achieved by posing network training as a constrained optimization problem where skip-connections are introduced during the early stages of training and subsequently phased out in a principled manner using Lagrange multipliers.

Throughout a series of experiments we demonstrate that the performance of VAN networks is stable, displaying a far smaller drop in performance as depth increases and thereby largely mitigating the degradation problem.

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

## A VAN RE-FORMULATION

The original formulation for the VAN residual block was as follows:

$$\mathbf{x}_{l+1} = \boldsymbol{\alpha}_l \cdot \mathcal{F}_l(\mathbf{x}_l, \mathbf{W}_l) + (1 - \boldsymbol{\alpha}_l) \cdot \mathbf{x}_l. \tag{9}$$

We thank an anonymous reviewer for suggesting that such a formulation may be detrimental to the performance of very deep VAN networks. The reason for this is that scaling constant within each block is always less than one, implying that the contributions of lower layers vanish exponentially as the depth increases. This argument is also provided in He et al. (2016b) who perform similar experiments with ResNets.

In order to validate this hypothesis, we compare the performance of VAN networks employing the residual block described in equation (4) and the residual block described in equation (9). The results, shown in Figure 4, provide evidence in favor of the proposed hypothesis. While both formulations for VAN networks obtain similar performances for shallow networks, as the depth of the network increases there is a more pronounced drop in the performance of VAN networks which employ residual blocks described in equation (9).

In a further experiment, we also studied the performance of ResNets with the following residual block:

$$\mathbf{x}_{l+1} = 0.5 \cdot \mathcal{F}_l(\mathbf{x}_l, \mathbf{W}_l) + 0.5 \cdot \mathbf{x}_l. \tag{10}$$

The results in Figure 4 demonstrate that ResNets which employ the residual blocks defined in equation (10) show a clear deterioration in performance as the depth of the network increases. Such a degradation in performance is not present when standard ResNets are employed.

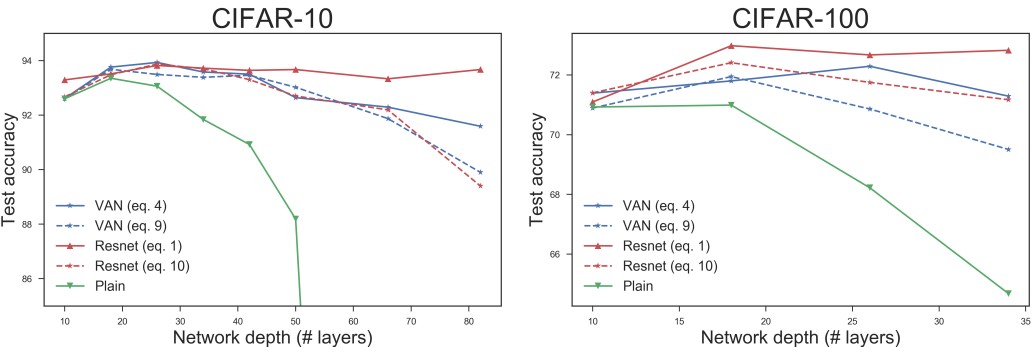

Figure 4: Results are shown VAN and ResNet networks with various different residual blocks. We note that the use of residual blocks with non-identity scaling coefficients leads to a larger drop in performance as the network depth increases. This drop is attributed to vanishing contributions from lower blocks (as all scalings are less than one).

# B    LAGRANGE MULTIPLIER TRACES

In this section we provide addition figures demonstrating the evolution of Lagrange multipliers, $\boldsymbol{\lambda}_l$ throughout training. We note that the updates to Lagrange multipliers are directly modulated by the current value of each $\boldsymbol{\alpha}_l$ (see equation (8)). As such, we also visualize the mean residuals of the $\boldsymbol{\alpha}_l$ parameters across all layers. This is defined as $\frac{1}{L} \sum_{l=1}^{L} (\mathbf{1} - \boldsymbol{\alpha}_l)^2$ and is a measure of the extent to which skip-connections are present in the network. Once all skip-connections have been removed, this residual will be zero and the values of Lagrange multipliers will no longer change.

This is precisely what we find in Figure 5. The left panel plots the mean value of Lagrange multipliers across all layers, while the right panel shows the mean residual of $\boldsymbol{\alpha}_l$ . We observe that for networks of different depths, once the constraint to remove skip-connections is satisfied, the value of Lagrange multipliers remains constant. This occurs at different times; sooner for more shallow networks whilst later on for deeper networks.

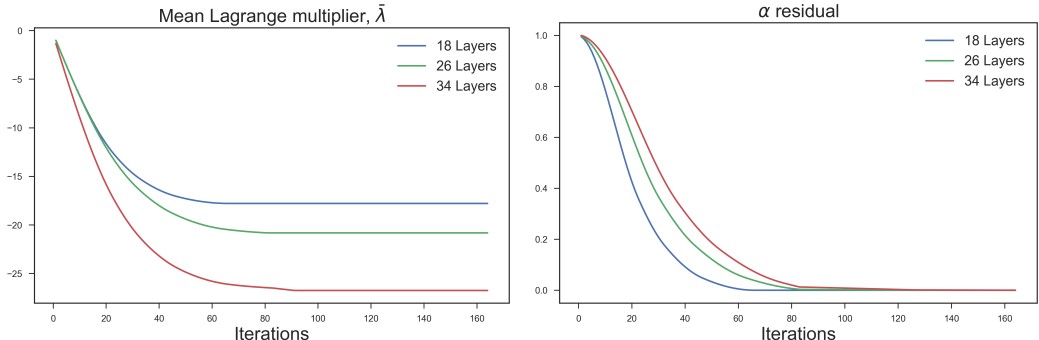

Figure 5: Lagrange multipliers, $\boldsymbol{\lambda}_l$ are shown on the left panel for networks of varying depth. After a certain number of iterations, the values of the Lagrange multiplier plateau as the constraint to remove skip-connections is satisfied. This results in no updates to the values of the Lagrange multipliers (see equation (8)). The right panel shows the mean $\boldsymbol{\alpha}_l$ residual. This residual directly modulates the magnitude of changes in Lagrange multipliers.

