# OpenReview forum: "Avoiding degradation in deep feed-forward networks by phasing out skip-connections"
_ICLR.cc/2018/Conference — Reject_

### Official Review · AnonReviewer2 · 2017-11-20
**Useful if somewhat marginal paper with issues.**

**Rating:** 6
**Confidence:** 4

**Review:**

EDIT: The rating has been changed. See thread below for explanation / further comments.

ORIGINAL REVIEW: In this paper, the authors present a new training strategy, VAN, for training very deep feed-forward networks without skip connections (henceforth called VDFFNWSC) by introducing skip connections early in training and then gradually removing them.

I think the fact that the authors demonstrate the viability of training VDFFNWSCs that could have, in principle, arbitrary nonlinearities and normalization layers, is somewhat valuable and as such I would generally be inclined towards acceptance, even though the potential impact of this paper is limited because the training strategy proposed is (by deep learning standards) relatively complicated, requires tuning two additional hyperparameters in the initial value of \lambda as well as the step size for updating \lambda, and seems to have no significant advantage over just using skip connections throughout training. So my rating based on the message of the paper would be 6/10.

However, there appear to be a range of issues. As long as those issues remain unresolved, my rating is at is but if those issues were resolved it could go up to a 6.

+++ Section 3.1 problems +++

- I think the toy example presented in section 3.1 is more confusing than it is helpful because the skip connection you introduce in the toy example is different from the skip connection you introduce in VANs. In the toy example, you add (1 - \alpha)wx whereas in the VANs you add (1 - \alpha)x. Therefore, the type of vanishing gradient that is observed when tanh saturates, which you combat in the toy model, is not actually combated at all in the VAN model. While it is true that skip connections combat vanishing gradients in certain situations, your example does not capture how this is achieved in VANs.
- The toy example seems to be an example where Lagrangian relaxation fails, not where it succeeds. Looking at figure 1, it appears that you start out with some alpha < 1 but then immediately alpha converges to 1, i.e. the skip connection is eliminated early in training, because wx is further away from y than tanh(wx). Most of the training takes place without the skip connection. In fact, after 10^4 iterations, training with and without skip connection seem to achieve the same error. It appears that introducing the skip connection was next to useless and the model failed to recognize the usefulness of the skip connection early in training.
- Regarding the optimization algorithm involving \alpha^* at the end of section 3: It looks to me like a hacky, unprincipled method with no guarantees that just happened to work in the particular example you studied. You motivate the choice of \alpha^* by wanting to maximize the reduction in the local linear approximation to \mathcal{C} induced by the update on w. However, this reduction grows to infinity the larger the update is. Does that mean that larger updates are always better? Clearly not. If we wanted to reduce the size of the objective according to the local linear approximation, why wouldn't we choose infinitely large step sizes? Hence, the motivation for the algorithm you present is invalid. Here is an example where this algorithm fails: consider the point (x,y,w,\alpha,\lambda) = (100, \sigma(100), 1.0001, 1, 1). Here, w has almost converged to its optimum w* = 1. Correspondingly, the derivative of C is a small negative value. However, \alpha* is actually 0, and this choice would catapult w far away from w*.

If I haven't made a mistake in my criticisms above, I strongly suggest removing section 3.1 entirely or replacing it with a completely new example that does not suffer from the above issues.

+++ ResNet scaling +++

There is a crucial difference between VANs and ResNets. In the VAN initial state (alpha = 0.5), both the residual path and the skip path are multiplied by 0.5 whereas for ResNet, neither is multiplied by 0.5. Because of this, the experimental results between the two architectures are incomparable.

In a question I posed earlier, you claimed that this scaling makes no difference when batch normalization is used. I disagree. Let's look at an example. Consider ResNet first. It can be written as x + r_1 + r_2 + .. + r_B, where r_b is the value computed by residual block b. Now let's assume we insert a scaling constant after each residual block, say c = 0.5. Then the result is c^{B}x + c^{B-1}r_1 + c^{B-2}r_2 + .. + r_B. Therefore, contributions of lower blocks vanish exponentially. This effect is not combated by batch normalization.

So the learning dynamics for VAN and ResNet are very different because of this scaling. Therefore, there is an open question: are the differences in results between VAN and ResNet in your experiments caused by the removal of skip connections during training or by this scaling? Without this information, the experiments have limited value. In fact, I suspect that the vanishing of the contribution of lower blocks bears more responsibility for the declining performance of VAN at higher depths than the removal of skip connections.

If my assessment of the situation is correct, I would like to ask you to repeat your experiments with the following two settings:

- ResNet where after each block you multiply the result of the addition by 0.5, i.e. x_{l+1} = 0.5\mathcal{F}(x_l) + 0.5x_l
- VAN with the following altered equation: x_{l+1} = \mathcal{F}(x_l) + (1-\alpha)x_l, i.e. please remove the alpha in front of \mathcal{F}. Also, initialize \alpha to zero. This ensures that VAN starts out as a regular ResNet.

+++ writing issues +++

Title:

- "VARIABLE ACTIVATION NETWORKS: A SIMPLE METHOD TO TRAIN DEEP FEED-FORWARD NETWORKS WITHOUT SKIP-CONNECTIONS" This title can be read in two different ways. (A) [Train] [deep feed-forward networks] [without skip-connections] and (B) [Train] [deep feed-forward networks without skip connections]. In (A), the `without skip-connections' modifies the `train' and suggests that training took place without skip connections. In (B), the `without skip-connections' modifies `deep feed-forward networks' and suggests that the network trained has no skip connections. You must mean (B), because (A) is false. Since it is not clear from reading the title whether (A) or (B) is true, please reword it.

Abstract:

- "Part of the success of ResNets has been attributed to improvements in the conditioning of the optimization problem (e.g., avoiding vanishing and shattered gradients). In this work we propose a simple method to extend these benefits to the context of deep networks without skip-connections." Again, this is ambiguous. To me, this sentence implies that you extend the benefit of avoiding vanishing and exploding gradients to fully-connected networks without skip connections. However, nowhere in your paper do you show that trained VANs have less exploding / vanishing gradients than fully-connected networks trained the old-fashioned way. Again, please reword or include evidence.
- "where the proposed method is shown to outperform many architectures without skip-connections" Again, this sentence makes no sense to me. It seems to imply that VAN has skip connections. But in the abstract you defined VAN as an architecture without skip connections. Please make this more clear.

Introduction:
- "Indeed, Zagoruyko & Komodakis (2016) demonstrate that it is better to increase the width of ResNets than the depth, suggesting that perhaps only a few layers are learning useful representations." Just because increasing width may be better than increasing depth does not mean that deep layers don't learn useful representations. In fact, the claim that deep layers don't learn useful representations is directly contradicted by the paper.

section 3.1:
- replace "to to" by "to" in the second line

section 4:
- "This may be a result of the ensemble nature of ResNets (Veit et al., 2016), which does not play a significant role until the depth of the network increases." The ensemble nature of ResNet is a drawback, not an advantage, because it causes a lack of high-order co-adaptataion of layers. Therefore, it cannot contribute positively to the performance or ResNet.

As mentioned in earlier comments, please reword / clarify your use of "activation function". It is generally used a synonym for "nonlinearity", so please use it in this way. Change your claim that VAN is equivalent to PReLU. Please include your description of how your method can be extended to networks which do allow for skip connections.

+++ Hyperparameters +++

Since the initial values of \lambda and \eta' are new hyperparameters, include the values you chose for them, explain how you arrived at those values and plot the curve of how \lambda evolves for at least some of the experiments.

---

> ### Author Response · Authors · 2017-12-29
> **Reply**
>
> We thank the reviewer for providing a thorough and thoughtful review. We have made many of the changes suggested in the review, leading to an improved manuscript in the process. Below we respond to each of the comments in turn.
>
> Toy problem:
> The reviewer raised important issues relating to the toy problem and its relevance to the proposed framework. As such, we have removed this motivating example from the updated manuscript.
>
> ResNet scaling:
> We thank the reviewer for alerting us to an important shortcoming of the proposed method. Following the reviewers suggestion, we have run experiments with the following residual block structure:
> 		\mathcal{F}(x_l, W_l) + (1 - \alpha_l) x_l
> Further, as suggested by the reviewer we have initialized with \alpha=0 resulting in networks that were equivalent to ResNets at initialization.
> Our experiments suggest that this formulation leads to improved results, especially in the context of very deep networks. Further, we have also run the ResNet scaling experiment suggested by the reviewer and find similar degradation in performance for deep networks. These results, which are reported in Appendix A, provide evidence for the reviewers hypothesis that modulating the \mathcal{F} by some  \alpha_l \neq 1 leads to a vanishing contribution from shallow blocks.
>
> Writing issues:
> We have the following changes:
>  - Title: As suggested by the reviewer, we have amended the title to clearly resemble the goal of the proposed method and our contribution. The new title is: “Avoiding degradation in deep feed-forward networks by phasing out skip-connections”.
>  - Abstract: The abstract has also been re-written in order to clarify the objectives and contributions of our work.
>  - Introduction, claim about relationship between width and depth: the paragraph in question has been removed as it was not relevant to the goals of the proposed method.
>  - Section 4, comment about ensemble nature of ResNets: Whether the ensemble nature of ResNets is an advantage of disadvantage is unclear. While boosting will typically lead to better performance, the reviewer notes that the ensemble nature of ResNets may actually be detrimental as it leads to co-adaptation of features. In order to avoid entering into this discussion we have removed this sentence.
>  - Use of activation function/non-linearity: this has been clarified throughout.
>
> Hyper-parameters:
> we have updated the manuscript to clearly state the choice of hyper-parameters. In particular, we now clearly state the choice of \eta’ and provide traces for the Lagrange multipliers in Appendix B.

---

> > ### Comment · AnonReviewer2 · 2018-01-02
> > **Rebuttal rebuttal (1/2)**
> >
> > Dear Authors,
> >
> > Thank you for your detailed response and updated manuscript. I changed my rating to 5, with the rating becoming a 6 if (A) you explain how you arrived at your hyperparameter choices and (B) if that explanation is reasonable, i.e. does not reveal weaknesses in statistical validity. Where did the numbers 165 / 82 / 125 come from in the third paragraph of section 4.2? How where VAN-specific hyperparameters (\eta' and initial \lambda) chosen?
> >
> > Also I just noticed that for CIFAR-10, you considered much deeper networks than for CIFAR-100. Why is that? Can you include results for, say, 80-layer nets in CIFAR-100?
> >
> > There seems to be something wrong with figure 2 (right graph). The gold curve seems to start at 0.25, but it should start at 1 under the new formulation of VAN where the \alpha are initialized to 0.
> >
> > I read the reviews written by the other reviewers and your rebuttal to them. While I do not presume to speak for the other reviewers, I think their concerns were addressed well in both the rebuttals and the revised version. However, I understand that if the other reviewers do not opt to increase their ratings in response to those rebuttals, it may not be worth your time to respond to this comment or upload another revision. I just wanted to let you know that I would not be offended if you did not respond to this comment or upload another revision.

---

> > > ### Author Response · Authors · 2018-01-03
> > > **Reply**
> > >
> > > Thank you for quick response - we have uploaded an updated manuscript incorporating the suggested changes. Please find our response to comments below.
> > >
> > > Choice of parameters:
> > > The choice of the training schedule and learning rate where taken directly from He et al., (2016). In this paper, the authors state: “We start with a learning rate of 0.1, divide it by 10 at 32k and 48k iterations, and terminate training at 64k iterations” where we take each iteration to refer to one batch of training. Based on a training set of 50k samples and a batch size of 128 this implies that each epoch corresponds to approximately 50k/128 = 390 iterations. Therefore 32k, 48k and 64k iterations correspond roughly to the 82, 125 and 165 epochs.
> > >
> > > Choice of VAN-specific hyper-parameters
> > > The choice of VAN hyper-parameters was based on a small number of experiments (as noted by AnonReviewer3, we did not run extensive hyper-parameter searches).
> > >
> > > CIFAR-100 experiments
> > > Our goal through the experiments was to demonstrate that degradation did not occur with VANs.  In the case of the CIFAR100 experiments, the degradation of plain networks is already clear for networks of depth 26 and 34 (we think  this may be due to the reduced number of training examples per class). As such, we did not run experiments for deeper architectures on CIFAR-100.
> > >
> > > Axis of right panel in Figures 2, 3
> > > We thank the reviewer for spotting this - this scaling was because we had accidentally divided by the number of residual blocks (this was fixed at 4 for all experiments). This has been corrected.
> > >
> > > Minor points
> > > We thank the reviewer for the suggestions, most of which have been incorporated into the updated manuscript.
> > >
> > >
> > > References:
> > > He, Kaiming, et al. "Deep residual learning for image recognition." Proceedings of the IEEE conference on computer vision and pattern recognition. 2016.

---

> > > > ### Comment · AnonReviewer2 · 2018-01-04
> > > > **ok**
> > > >
> > > > ok

---

> > ### Comment · AnonReviewer2 · 2018-01-02
> > **Rebuttal rebuttal (2/2)**
> >
> >
> > Some minor points I noticed while reading the updated manuscript:
> >
> > - "...is shown to greatly decrease the degradation effect (compared to plain networks) and is often competitive with ResNets." This sentence suggests that a VAN is not considered by the authors to be a "plain network", but a seperate architecture. "However, the degradation problem persists in the context of plain feed-forward networks. In this work we propose a simple method to address this issue." This part suggests that VAN's are considered by the authors to be a "plain network", and VAN to be merely an optimization "method". I suggests the authors decide whether they want to frame VAN as an architecture seperate from ResNet, HighwayNet and plain networks, or as an optimization method for plain networks and then use this framing throughout the paper.
> > - "even within training data". replace this with "even when the network is applied to training data"
> > - "ResNets can be considered a special case of Highway Networks" not in the way you defiend ResNet and Highway networks. (1) is not a special case of (2) as the highway network cannot represent the skip path's multiplication with W'.
> > - While I'm flattered by the inclusion of footnote 2, I don't think it is necessary / appropriate to acknowledge me in this way. In theory, reviewers are supposed to be helpful.
> > - "In all networks the first layer was a fully connected plain layer followed by l layers and a final softmax layer." What do you mean by "l layers"? Do you mean "l residual blocks"? Please clarify.
> > - "In the case of ResNets and VAN, the residual function consisted of batch-normalization followed by ReLU." This sounds like it does not include a linear transformation, but I assume it does. Also, is the linear transformation applied before or after those two operations? Please clarify.
> > - replace "gradual removed" with "gradual removal"
> > - "This provides evidence that the gradual removed of skip-connections via Lagrange multipliers leads to improved generalization performance" compared to what?
> > - "Finally, we note that VAN networks  obtain  competitive  results  across  all  depths.   This  is  particularly  true  in  the  context  of shallow networks (l ≤ 10 layers) where VAN networks obtain competitive test accuracies." You seem to contradict yourself here. First you say that VAN is completitive at all depths and then you insinuate that it is only competitive for l <= 10.
> > - "Crucially, we note that VAN networks outperform plain networks in this context, " It looks to me that VAN outperforms plain networks in all contexts, i.e. for all values of l.
> > - "As a more realistic benchmark we consider the CIFAR-10 and CIFAR-100 datasets." It sounds like you are declaring MNIST / Fashion-MNIST "unrealistic". I don't think this is necessary as I don't think CIFAR is more realistic than MNIST. Digit recognition is a valid field of application for deep networks.
> > - "with a fully connected softmax layer" Do you mean fully-connected layer FOLLOWED BY a softmax layer or do you mean JUST a softmax layer which you describe as fully-connected? Please clarify.
> > - in table 2, please also include the more recent benchmarks wide resnet and densenet
> > - "This manuscript presents a simple method for training deep feed-forward networks without suffering from the degradation problem." Doesn't it still suffer from the degradation problem, albeit to a lesser extent?
> > - "Throughout a series of experiments we demonstrate that the performance of VAN networks is stable as network depth increases" really? I thought it degrades slightly. That does not seem to me to be the same as being stable.
> > - I don't think you need to include Appendix A, unless you are making an argument that the alternative formulation with the 0.5's has value independent of the formulation used in the main paper. I don't think you are trying to make that case. In my original review, while I suggested you run the experiments shown in Appendix A, this was more for my and your benefit in terms of understanding what's going on. I don't think the 0.5 formulation needs to appear in the paper.
> > - replace "by the currently value" by "by the current value"
> > - fix grammar in "The left panel plots the mean Lagrange multiplier, λl are shown for networks of varying depth."

---

### Official Review · AnonReviewer1 · 2017-11-26
**Simple idea, not explored thoroughly enough**

**Rating:** 6
**Confidence:** 4

**Review:**

UPDATED COMMENT
I've improved my score to 6 to reflect the authors' revisions to the paper and their response to my and R2's comments. I still think the work is somewhat incremental, but they have done a good job of exploring the idea (which is nice).

ORIGINAL REVIEW BELOW

The paper introduces an architecture that linearly interpolates between ResNets and vanilla deep nets (without skip connections). The skip connections are penalized by Lagrange multipliers that are gradually phased out during training. The resulting architecture outperforms vanilla deep nets and sometimes approaches the performance of ResNets.

It’s a nice, simple idea. However, I don’t think it’s sufficient for acceptance. Unfortunately, this seems to be a simple idea that doesn't work as well as the simpler idea (ResNets) that inspired it. Moreover, the experiments are weak in two senses: (i) there are lots of obvious open questions that should have been explored and closed, see below, and (ii) the results just aren’t that good.

Comments:

1. Why force the Lag. multipliers to 1 at the end of training? It seems easy enough to treat the alphas as just more parameters to optimize with gradient descent. I would expect the resulting architecture to perform at least as well as variable action nets. If not, I’d be curious as to why.

2.Similarly, it’s not obvious that initializing the multipliers at 0.5 is the best choice. The “looks linear” initialization proposed in “The shattered gradients problem” (Balduzzi et al) implies that alpha=0 may work better. Did the authors try any values besides 0.5?

3. The final paragraph of the paper discusses extending the approach to architectures with skip-connections. Firstly, it’s not clear to me what this would add, since the method is already interpolating in some sense between vanilla and resnets. Secondly, why not just do it?

---

> ### Author Response · Authors · 2017-12-29
> **Reply**
>
> We thank the reviewer for raising important issues. We respond to each below.
>
> Not enforcing constraint on alphas:
> We have run the additional experiments suggested by the reviewer. As expected, when no constraint is enforced on the alphas (this corresponds to setting the Lagrange multipliers to 0 in equation (6)), the performance of VAN networks does indeed improve. These additional experiments have been included in the updated version of the manuscript (see Figures 2 and 3 as well as Table 2).
>
> Initialization of alpha:
> Our original experiments focused on the choice of alpha=0.5 at initialization. However, the reviewer correctly notes that such an initialization will not necessarily be optimal. Following the suggestion of AnonReview2 we have reformulated the VAN equation as follows:
> 		\mathcal{F}(x_l) + (1 - \alpha_l) x_l
> And initialize with alpha_l=0. This ensures that VANs are equivalent to ResNets at initialization (while this was not previously the case). More importantly, such a reparameterization leads to improved performance - an empirical comparison is provided in Appendix A.
>
> VANs with skip-connections:
> the reviewer correctly notes that it is intuitively obvious how to combine VAN residual blocks with architectures that contain skip-connections. One way the two could be combined would be by not enforcing the constraint on \alpha_l in VANs (as we have done in response to a previous comment). We have removed this sentence from the manuscript.

---

### Official Review · AnonReviewer3 · 2017-11-27
**Interesting idea but suffers from lack of motivation and sound experiments (UPDATED)**

**Rating:** 6
**Confidence:** 5

**Review:**

Update (original review below):
The authors have addressed several of the reviewers' comments and improved the paper.
The motivation has certainly been clarified, but in my opinion it is still hazy. The paper does use skip connections, but the difference is that they are phased out over training. So I think that the motivation behind introducing this specific difference should be clear. Is it to save the additional (small) overhead of using skip connections?
Nevertheless, the additional experiments and clarifications are very welcome.

For the newly added case of VAN(lambda=0), please note the strong similarity to https://arxiv.org/abs/1611.01260 (ICLR2017 reviews at https://openreview.net/forum?id=Sywh5KYex). In that report \alpha_l is a scalar instead of a vector.

Although it is interesting, the above case case also calls into question the additional value brought by the use of constrained optimization, a main contribution of the paper.

In light of the above, I have increased my score since I find this to be an interesting approach, but in my opinion the significance of the results as they stand is low. The paper demonstrates that it is possible to obtain very deep plain networks (without skip connections) with improved performance  through the use of constrained optimization that gradually removes skip connections, but the value of this demonstration is unclear because a) consistent improvements over past work or the \lambda=0 case were not found, and b) The technique still relies on skip connections in a sense so it's not clear that it suggests a truly different method of addressing the degradation problem.

Original Review
=============
Summary:
The contribution of this paper is a method for training deep networks such that skip connections are present at initialization, but gradually removed during training, resulting in a final network without any skip connections.
The paper first proposes an approach based on a formulation of deep networks with (non-parameterized, non-gated) skip connections with an equality constraint that effectively removes the skip connections when satisfied. It is proposed to optimize the formulation using the method of Lagrange multipliers.
A toy model with a single unit is used to illustrate the basic ideas behind the method. Finally, experimental results for the task of image classification are reported using the MNIST, Fashion-MNIST, and CIFAR datasets.

Quality and significance:
The proposed methodology is simple and straightforward. The analysis with the toy network is interesting and helps illustrate the method. However, my main concerns with this paper are related to motivation and experiments.

The motivation of the work is not clear at all. The stated goal is to address some of the issues related to the role of depth in deep networks, but I think it should be clarified which specific issues in particular are relevant to this method and how they are addressed. One could additionally consider that removing the skip connections at the end of training reduces the computational expense (slightly), but beyond that the expected utility of this investigation is very hazy from the description in the paper.

For MNIST and MNIST-Fashion experiments, the motivation is mentioned to be similar to Srivastava et al. (2015), but in that study the corresponding experiment was designed to test if deeper networks could be optimized. Here, the generalization error is measured instead, which is heavily influenced by regularization. Moreover, only some architectures appear to employ batch normalization, which is a potent regularizer. The general difference between plain and non-plain networks is very likely due to optimization difficulties alone, and due to the above issues further comparisons can not be made from the results.

For the CIFAR experiments, the experiment design is reasonable for a general comparison. Similar experimental setups have been used in previous papers to report that a proposed method can achieve good results, but there is no doubt that this does not make a rigorous comparison without employing expensive hyper-parameter searches. This is not the fault of the present paper but an unfortunate tradition in the field. Nevertheless, it is important to note that direct comparison should not be made among approaches with key differences. For the reported results, Fitnets and Highway Networks did not use Batch Normalization (which is a powerful regularizer) while VANs and Resnets do. Moreover, it is important to report the training performance of deeper VANs (which have a worse generalization error) to clarify if the VANs suffered difficulties in optimization or generalization.

Clarity:
The paper is generally well-written and easy to read. There are some clarity issues related to the use of the term "activation function" and a typo in an equation but the authors are already aware of these.

---

> ### Author Response · Authors · 2017-12-29
> **Reply**
>
> We thank the reviewer for raising three important points, which we respond to below
>
> Motivation:
> We agree with the reviewer that the motivation of the proposed work was unclear in the original submission. As a result, we have updated the manuscript to clearly state out motivation: which is to address the degradation problem in deep feed-forward networks. While ResNets and Highway networks address this issue by introducing skip-connections or gating mechanisms, we look to tackle the problem from the perspective of constrained optimization. As such, the objective of the our work is to propose a new training regime for plain networks which explicitly addresses the issue of performance degradation with depth for plain feed-forward networks.
>
> In order to clearly reflect our motivation and contribution we have amended the title of the manuscript as well as clarified the abstract and the introduction.
>
>
> Regarding MNIST and MNIST-Fashion experiments:
> Batch-normalization was employed across all architectures in the MNIST experiments (although we do acknowledge that the original Highway network experiments did not use batch norm). We have updated the manuscript to clearly state this. As a result, we do believe that comparisons across the different architectures are valid on this experiment.
> While Srivastava et al. (2015) reported the training cross-entropy,  it is unclear to us why reporting the generalization performance does not serve as an indication of successful optimization across various networks given that all networks employed batch normalization.
>
> CIFAR experiments:
> The reviewer correctly notes that extensive hyper-parameter searches where not run for the CIFAR experiments. This is because the objective of the experiments in Section 4 was not to achieve state-of-the-art performance but rather to demonstrate that VAN networks do not suffer from the degradation problem to the same extent as plain feed-forward networks. We feel that these experiments serve to validate this claim.
> Regarding comparisons with alternative architectures (e.g., FitNets and Highway networks) we have amended Table 2 to state which architectures did and did not use batch-normalization.

---

> ### Author Response · Authors · 2018-01-16
> **Reply to update**
>
> Thank you for your feedback as well as for increasing the score of the paper. We respond to the remaining comments below.
>
> Similarity to Savarese et al., (2016)
> We thank the reviewer for suggesting this paper, which we will reference in an updated manuscript. As noted by the reviewer, the similarity between our work and that of Savarese et al., (2016) occurs when \lambda=0. This corresponds to the scenario where no constraints are not enforced on \alpha (that is, skip-connections are not removed). The comparison to the \lambda=0 case was included in order to study the capacity and flexibility of VAN networks without the need to satisfy the constraint to remove skip-connections. However, our main focus consisted in removing skip-connections (i.e., case where \lambda \neq 0), which is different from Savarese et al., (2016).
>
> Contributions and motivation
> We thank the reviewer for noting that the motivation has been clarified. In an updated manuscript we will further clarify that the motivation for this work is not to reduce the computational cost associated with skip-connections but instead to pose the learning of deep networks in the context of constrained optimization. The motivation behind the proposed method is to introduce skip-connections penalized by Lagrange multipliers into the architecture of our network. In this manner, skip-connections play an important role during the initial training of the network (e.g., by avoiding shattered gradients) and are subsequently removed in a principled manner. As noted by the reviewer, such an approach allows us to train deep plain networks (which do not contain skip-connections, as these are ultimately removed) without suffering degradation to the same extent as when ordinary training is employed.

---

### Comment · AnonReviewer2 · 2017-11-09
**Lagrangian clarification**

Dear authors, I am one of the reviewers and would like to ask a couple of question. In equation (6), you show the Lagrangian of the constrained problem. Is there a deeper reason why you do this? Or are you simply using the term in equation (6) as your objective function and the fact that it is the Lagrangian is merely an illustration? The reason I ask is that it's been a while since I took Optimization 101 :), meaning I would have to read up on the Lagrangian to remind myself of the associated theory. If you let me know that understanding Lagrangian optimization is necessary for the paper, I will go and do that. If not, I won't.

Also, there seems to be something wrong with + and - signs throughout section 3. If you take the derivative of (6) with respect to \alpha and \lambda, one does not get out (7) and (8). It seems that in (7) the second minus should be a plus and in (8), the first plus should be a minus. The exact same problem occurs at the bottom of page 4 for the toy example.

Also, to what value do you initialize \lambda? For equation (6) to make sense, \lambda has to be initialized to negative values so that the term that is added to \mathcal{C} is positive, and it needs to be positive as it is a penalty ...

Also, you say " In the case of VAN networks the α values were initialized to 0.5 · 1 for all layers. As such, during the initial stages of training VAN networks had the same activation function as ResNets." I don't understand this. In the initial state, the weight of the residual path and skip path for ResNet is both 1. But in VAN, the weight of the residual path and the skip path are both 0.5. Hence, as far as I can tell, they are *not* the same. Also, how did you initialize \alpha for the CIFAR experiments?

Also, what do you mean by "activation function". Do you mean \mathcal{F} or do you mean just the nonlinearity? You seem to be using \mathcal{F} for both, which is confusing.

---

> ### Author Response · Authors · 2017-11-15
> **Reply**
>
> Dear Reviewer,
>
> Thank you for your questions. We respond to each of them below.
>
> 1) Use of Lagrange multipliers
>
> The objective of the proposed method is to extend some of the benefits associated with skip-connections to the context of networks without skip-connections. We propose to do this by working in the framework of constrained optimisation. As such, the use of Lagrange multipliers is not an illustration but a tool we employ in order to solve equation (5), which is the true objective. Equation (6) is the associated Lagrangian, which combines the objective in equation (5) with a term enforcing the constraint. Our updates then seek a saddle point of this Lagrangian, by minimizing it with respect to the parameters W and \alpha, and maximizing with respect to the Lagrange multipliers \lambda.
>
> 2) + and - signs in equations (7) and (8):
>
> As stated above, the updates should minimize the Lagrangian with respect to \alpha and W, while maximizing it with respect to \lambda. We thank the reviewer for pointing out the mistake in equation (7). It should read as:
>  \alpha_l \leftarrow  \alpha_l - \eta  ( \frac{\partial C}{ \partial \alpha_l } + \ \lambda_l  )
>
> However, with respect to equation (8), the plus sign is indeed correct. This is because we wish to maximise with respect to the Lagrange multipliers (in order to enforce the equality constraint with increasing severity). The signs in the equations in the section on the toy model are also correct; the reason they look different is because the constraint has been (equivalently) encoded as 1 - \alpha there, rather than as \alpha - 1 in equation (6).
>
> 3) Initialization of \lambda
>
> Throughout our experiments we initialized \lambda to -1. Because we enforced that \alpha remain in [0,1], all updates to \lambda were <=0. These two facts ensured that \lambda remained negative. An updated version of the manuscript will reflect this. However, as noted in the paragraph after equation (8), the nature of the \lambda updates dictates that \lambda will be monotonically decreasing. Thus even if we initialized \lambda to be positive, subsequent updates would push it towards negative values.
>
> 4) Relation between VAN at \alpha=.5 and ResNets
>
> The reviewer is correct in noting that when \alpha=.5 there is a similarity between ResNets and VANs, but the activation is not exactly the same (as the VAN activation is scaled by 0.5). We will correct our language to reflect this. However, what we intended to highlight was that the balance between non-linearity and skip-connection was equal (as in a ResNet) whereas for arbitrary \alpha, equation (4) defines a sum where the non-linearity and the skip-connection may not necessarily receive equal weights.  Further, when batch normalisation is employed the activation functions are effectively the same as the effects of scaling are removed.
>
> 5) Initialization of \alpha for CIFAR experiments
>
> We initialized \alpha=0.5 - see the final paragraph on page 6. This initialization was used throughout all experiments; we will make this clear in the updated manuscript.
>
> 6) Activation function and non-linearity.
>
> We agree with the reviewer that our language regarding activation function and non-linearity could be more clear. When we say "activation function" we refer to \mathcal{F}. In the experiments section we make statements such as "the ReLU activation function ..." when we should say "the ReLU non-linearity". We will correct this in the updated version of the manuscript.
>
>
> Thanks for you questions - we hope to have clarified them. If you have any more questions (or need further details) please let us know :)

---

### Comment · AnonReviewer2 · 2017-11-16
**More questions**

Thank you for your responses so far. I need to ask a few more question before writing my final review.

"Throughout the experiments we will often take the nonlinearity in Fl to be ReLU in which case the activation function corresponds to the parametric ReLU, for which the term @C can be efficiently computed (He et al., 2015)."

Firstly, please try to avoid using 'activation function' to refer to \mathcal{F} as it is generally considered to be a synonym for 'nonlinearity'. However, if you do use 'activation function' to refer to \mathcal{F}, then the statement "the activation function corresponds to the parametric ReLU" makes no sense, because parametric ReLU is a nonlinearity. I take it \mathcal{F} is not just composed of a nonlienarity, but also of convolutional and / or normalization layers.

However, even if I take the statement "the activation function corresponds to the parametric ReLU" to mean "\mathcal{F} corresponds to a sequence of operations, the last of which is parametric ReLU", it still makes no sense because the parameter alpha does not impact \mathcal{F}. In equation (4), it is only applied after \mathcal{F} is computed. So I take it you use 'activation function' to refer to function that computes x_{l+1} from x_l in this instance? Again, this is ambiguous.

But even if we take 'activation function' to mean 'the function that  computes x_{l+1} from x_l' and thus "the activation function corresponds to the parametric ReLU" to mean "the function that  computes x_{l+1} from x_l  corresponds to a sequence of operations, the last of which is parametric ReLU", it is still false if equation (4) is true. Let's assume that \mathcal{F} = ReLU(Wx) for some W. Then we have x_{l+1} = (1-\alpha)*x_l + \alpha*ReLU(Wx_l) according to (4). But this is not the same as PReLU(Wx_l). In fact, we have PReLU(Wx_l) = (1-\alpha)*Wx_l + \alpha*ReLU(Wx_l).

Please clarify.


"However, the proposed method is quite general and can be readily extended to networks which do allow for skip-connections."


How?




Thanks,

---

> ### Author Response · Authors · 2017-11-16
> **Reply**
>
> Thank you for the further questions, please see our replies below.
>
> 1) Relation to parametric ReLU
>
> We agree that our language is ambiguous and will update the manuscript to avoid conflating 'activation function' with \mathcal{F} (note that currently we are unable to upload a new version).
>
> Regarding the various interpretations of line 4 p4, the reviewers final interpretation is correct and we agree that there is only a correspondence between the proposed method and parametric ReLUs when W is the identity. Our intention was to highlight a similarity with the parametric ReLU. However, we note that our claims regarding the gradient still holds; the gradient with respect to \alpha will be simple and computationally cheap.
>
> 2) Extension to networks with skip-connections
>
> One possible extension would be to enforce the constraint in equation (5) with an inequality where we constrain the total budget on \alpha_l across all layers.  This would lead to skip-connections being removed across some layers and retained across others, providing insights into where in a network skip-connections are most important.

---

### Author Response · Authors · 2018-01-05
**New Revision**

We thank the reviewers for all their helpful comments. We have now posted an updated manuscript incorporating the comments of reviewers. We have responded to each reviewer below. We highlight the main changes to the manuscript here:

Experiments:
We have run additional experiments as suggested by reviewers. In particular, we have run experiments without enforcing the equality constraint is not enforced on \alpha (as requested by AnonReviewer1). This comparison is included in order to study the capacity and flexibility of VAN networks without the constraint to remove skip-connections. Following suggestions from AnonReviewer2 we have also reformulated the residual block structure of the proposed method, leading to improved results. This reformulated also provides a clear choice for the initialization of \alpha values, which was a concern for AnonReviewer1 (we now initialize to \alpha, ensuring that VANs are equivalent to ResNets at initialization).

Motivation:
The motivation for the proposed work was unclear in the original submission. As a result, we have updated the manuscript to clearly state out motivation: which is to address the degradation problem in deep feed-forward networks. While ResNets and Highway networks address this issue by introducing skip-connections or gating mechanisms, we look to tackle the problem from the perspective of constrained optimization. As such, the objective of the our work is to propose a new training regime for plain networks which explicitly addresses the issue of performance degradation with depth for plain feed-forward networks.

---

> ### Public Comment · ~Xiangyu_He1 · 2020-03-05
> **Nice work with great practical significance**
>
> Deep feed-forward networks without skip-connections can be simplified as "Convolution+ReLU" pairs, which significantly contribute to the efficient deployment on CPU devices (both runtime-speed and memory efficiency). Though not accepted by ICLR, no doubt it is a nice job.

---

### Decision · Program_Chairs · 2018-01-29
**ICLR 2018 Conference Acceptance Decision**

**Decision:**

Reject

**Comment:**

Pros:
+ Interesting perspective on training deep networks

Cons:
- Not a lot of practical significance: why would one want to use this algorithm over standard methods like ResNets or highway networks given that the proposed algorithm is more complex than established methods?